# Mitigation of the Adverse Effects of the El Niño (El Niño, La Niña) Southern Oscillation (ENSO) Phenomenon and the Most Important Diseases in Avocado cv. Hass Crops

**DOI:** 10.3390/plants9060790

**Published:** 2020-06-24

**Authors:** Joaquín Guillermo Ramírez-Gil, Juan Camilo Henao-Rojas, Juan Gonzalo Morales-Osorio

**Affiliations:** 1Departamento de Agronomía, Facultad de Ciencias Agrarias, sede Bogotá, Universidad Nacional de Colombia, Bogotá DC 111321, Colombia; 2Corporación Colombiana de Investigación Agropecuaria-Agrosavia, Centro de Investigación La Selva-Rionegro, Ríonegro-Llanograde 250047, Colombia; jhenao@agrosavia.co; 3Departamento de Ciencias Agronómicas, Facultad de Ciencias Agrarias, sede Medellín, Universidad Nacional de Colombia, Medellín 0550034, Colombia; jgmoralesos@unal.edu.co

**Keywords:** native genotypes, sustainable management, climatic variability, stress reduction

## Abstract

Areas cultivated with Hass avocado crops in Colombia have growth rapidly. One of the major limitations is the avocado wilt complex disease (AWC) caused by biotic and abiotic factors which have increased under the El Niño southern oscillation ENSO phenomenon (El Niño, La Niña). The objective of this study was to evaluate different strategies for mitigating the adverse effects associated with the ENSO phenomenon and AWC in avocado crops. We evaluated native materials, mulches, and parameters associated with the production of seedlings and planting practices in the field. The response variables tested were plant development, incidence, severity, mortality, and microbial dynamics, among others. The results indicated that native genotypes of *Persea americana* had different levels of adaptability to drought and flooding conditions. These genotypes also showed some degree of resistance to *Phytophthora cinnamomi* and *Verticillium* sp. infection with several degrees of rootstock-scion incompatibility with the Hass cultivar. In addition, mulch reduced the variability of soil moisture and temperature in the soil profile. Adequate selection of genotypes and new tools for planting have decreased the susceptibility to adverse effects associated with the ENSO phenomenon and the incidence and mortality caused by diseases under drought and flooding conditions. This work presents alternatives to mitigate adverse effects of climate variability in avocado crops under tropical conditions.

## 1. Introduction

Extreme variations in edaphoclimatic parameters such as temperature, CO_2_ levels, precipitation, amongst others, has been associated with crop vulnerability [1,2,3]. Such extreme variations have been observed during climate change events such as the El Niño Southern Oscillation phenomenon (ENSO). Climate change can impact global agricultural productivity through a wide range of processes that affect sustainability and can increase risks to global food security [1,2,3,4]. The potential impacts of this phenomenon can be more visible in tropical regions [4,5]. Weather can be modified radically under climate change scenarios, and their effects on plant species can be different according to their genetic composition, geographical distribution, population dynamics, symbiotic relationships, and many others [6,7,8].

Colombia is a country that naturally presents a high climate variability, given its topographic and geographic conditions and the different phenomena that influence the territory [9]. In Colombia, analyses have been carried out looking for signs of climate change in long series of hydrological and climatic variables (40–45 years), which have confirmed the presence of increasing trends in variables such as temperature, relative humidity, evaporation, and precipitation [10,11,12]. In addition, the changes in precipitation and temperature that can occur during the ENSO phenomenon can have strong and negative socioeconomic implications for the country [13].

The ENSO phenomenon is defined as a spontaneous event typical of the tropical Pacific through interactions between the ocean and the atmosphere, which involve alternating cycles of warm “El Niño” and cold “La Niña” temperatures that drastically affects global climate. These phenomena occur intermittently every two to seven years [14,15]. There have been multiple negative consequences of the ENSO phenomenon in agriculture, highlighted by a reduction of yield and an increase in pests and diseases [16,17,18].

Hass avocado (*Persea Americana* Mill) is the most commonly planted commercial variety for the internal and export markets in the world. This genotype originated spontaneously and is mainly associated with the Guatemalan and Mexican avocado races [19,20,21]. The Americas are home to 60% of the crops of this fruit. Mexico is the top producer with 34.5% of the global production; however, there are other important production areas in Colombia, Chile, Peru, Dominican Republic, and the United States (California and Florida) [22,23,24]. In addition, adequate development of avocado, as for any other plant species, is determined by its genotype and the interaction with the environment. Environmental variables such as temperature, precipitation, and others are crucial for determining the optimal zones for avocado production [24,25,26]. The ENSO phenomenon often affects seasonal temperature and precipitation [16], which can cause high economic impacts on avocado crops.

The avocado production system is perennial and presents an important number of technical limitations, from which root rot (RR) or avocado wilt complex (AWC) stand out due to its economic importance and worldwide distribution [27,28,29]. This pathology can be caused by biotic and abiotic factors, and each one can induce similar specific symptoms in the host. Usual symptoms in avocado plants are foliar yellowing, tissue flaccidity, growth retardation, excessive flowering and fructification in adult trees, defoliation, dieback, root rot, and plant death [30,31]. Avocado roots can be seriously affected by AWC causing large losses, usually between 45–90%, but which could reach 100% if no prompt and appropriate control measures are performed during all stages of crop establishment, growth, and development [27,29,32]. Disease development is considered to be influenced by high rainfall and soil waterlogging [18,33], which is not only associated with water dispersion of infective units but also with increased aggressiveness of microbial pathogens such as *Phytophthora cinnamomi* Rands [34,35].

Avocado plants are sensitive to extreme edaphoclimatic conditions such as those observed during the ENSO phenomenon that are the result of high rainfall and flooding, and other extreme severe droughts [18]. Avocado plants can decrease or halt growth and development during severe stress conditions causing large losses and delayed production [18,36,37]. Mitigation measures have been proposed for several crops and target negative impacts caused by climatic extremes such as experienced during the ENSO phenomenon [1].

The effects of organic matter on soil biogeochemical properties have been extensively studied, indicating complex interactions among microbiota, organic matter sources, as well as physical and chemical soil properties. Organic mulches have been shown to improve soil structure and aeration, promote beneficial microbial communities, excess water drainage, and at the same time retention of water available for plants, nutrient availability, and plant uptake through a number of mutualist symbiosis [38,39]. Roostock/scion combination can influence agronomic performance of avocado crops [36]. Avocado growth and development, tree size and vigor, fruit yield, and adaptation to adverse environments caused by biotic and abiotic stresses are frequently affected by rootstock selection [36,40,41]. Combined stresses can act synergistically to reduce crop production such as root rot, poor aeration, and waterlogging. Therefore, rootstock selection should consider adaptation to several combined stresses which suggests that local edaphoclimatic conditions require specific tests of rootstock selection [40,41]. In Colombia, research on mitigation measures for climate change in avocado crops is very poor and urgently needed.

The new trend in climate change studies worldwide is to perform modeling, risk analysis, and prediction, but also to move forward and look for alternatives that allow mitigation and adaptation of agricultural systems to extreme conditions of climate variability such as those observed during the ENSO phenomenon [1,2,42]. There is a direct relationship between soil moisture and AWC [18] and global evidence on variations of climatic parameters as a result of climate change. Those extreme variations of climatic parameters have been impacting areas planted with avocado crops in Colombia [24]. Considering the low amount of information on adaptation practices for avocado crops to climatic variability, one of the greatest challenges of avocado cultivation is the search for alternatives to adapt trees to the adverse effects of extreme climate variability in producing areas. The objectives of this study were to evaluate different strategies that could help to mitigate the adverse effects of climate variability on avocado crops and, at the same time, to evaluate their consequences on the avocado wilt complex disease.

## 2. Results

### 2.1. Adaptability of Native Rootstock Genotypes to Drought, Waterlogging, Tolerance to P. cinamomi and Verticillium sp. and Grafting Compatibility under Net House Condition

Native avocado genotypes that have been tested have shown different degrees of adaptability to variable soil moisture levels simulating drought or waterlogging (*p* < 0.05). Genotypes of the Guatemalan-type race (*Persea americana* var. *guatemalensis*) exhibited a better performance to drought (40, 30, 20, and 10%, of the maximum soil moisture retention capacity), followed by genotypes of the Mexican-type race (*Persea americana* var. *drymifolia*) and cv. Hass. Meanwhile, genotypes of the West Indian-type race (*Persea americana* var. *americana*) showed the lowest performance evidenced as a low relative rate increase (RRI) (g day^−1^) value (Table 1).

Regarding high soil moisture (90, 110, 130, and 150% of the maximum soil moisture retention capacity), the West Indian-type race genotypes were the best adapted (*p* < 0.05), followed by the genotypes of the Guatemalan-type race. For the area under the disease progress curve (AUDPC) values obtained by hypoxia/anoxia conditions, the same genotypes, i.e., West Indian-type race followed by Guatemalan-type race, showed the lowest disease values. The less adapted genotypes for high soil moisture and AUDPC values were those of the Mexican-type race, even with a lower adaptation than the control (Table 1).

The highest dry weight gain of avocado plants represented as the RRI values (*p* < 0.05) in the different materials was observed for the Guatemalan-type race, followed by the West Indian-type race and the Mexican race, that showed the lowest values. In general, the behavior of the avocado genotypes varied according to the soil moisture content; being the highest plant growth value at 50% of humidity for avocado genotypes of the Guatemalan- and Mexican-type race, whereas for the West Indian-type race the highest plant growth value was recorded at 90% of the maximum soil moisture capacity (Table 1). The AUDPC values were between 40 and 60% lower in the tested native genotypes of avocado than in the susceptible control of cv. Hass. The lowest AUDPC value in plants infected with *P. cinnamomi* was found for two genotypes of the West Indian-type race (AX1 and AX2) (*p* < 0.05), followed by genotypes GX3, GX4, and AX3, which are of the Guatemalan- and West Indian-type race, respectively. Remaining genotypes did not show significant differences as compared with the susceptible control of cv. Hass (Figure 1A). For the fungal pathogen *Verticillium* sp., only the Mexican-type race genotype had an AUDPC value between 25 and 40% lower than the susceptible control cv. Hass (*p* < 0.05) (Figure 1A).

For the variable graft compatibility, higher RRI values (*p* < 0.05) and no malformations in the grafted region were observed when cv. Hass scions were grafted onto any Guatemalan-type race genotype, with no significant differences observed as compared with the control of cv. Hass grafted on cv. Hass (Figure 1A).

Values obtained for different variables measured from all tested genotypes were analyzed by a grouping method to find relationships among the genotypes. The cluster analysis revealed the following four groups: In the first group (TD and GC encircled in blue in Figure 1B), there were the Guatemalan-type genotypes with the best results of adaptation to drought, high graft compatibility, medium to low resistance to *P. cinnamomi*, *Verticillium* sp., and hypoxia/anoxia; in the second group (RPC and TW encircled in red in Figure 1B), we identified together the West Indian-type race genotypes AX1, AX2, and AX4, which showed higher resistance to *P. cinnamomi* and hypoxia/anoxia, medium tolerance to high soil moisture and/or drought, medium graft compatibility and low resistance to *Verticillium* sp.; in the third group (HRD encircled in light blue in Figure 1B), we found together the West Indian-type race genotypes AX3 and AX5, which exhibited appropriate growth and development under high soil moisture, medium to low resistance to *P. cinnamomi*, hypoxia/anoxia and *Verticillium* sp., low tolerance to drought and low grafting compatibility with cv. Hass; the Mexican-type race genotypes were found together in the fourth group (RV encircled in black in Figure 1B) and they showed the highest resistance to the fungal pathogen *Verticillium* sp. (Figure 1B). 

### 2.2. Effect of Organic Mulch Addition on Avocados Roots Growth, Microbial Populations, and Reduction of Stress Associated with Environmental Conditions under Field Conditions

The effect of mulch addition on several variables of interest in avocado crops such as root quantity and quality, microbial populations, and soil moisture were measured in field conditions. The analysis revealed that the addiction of organic mulch induced root production (*p* < 0.05) with the best results observed for the addition of chopped plants and mushroom residues. Root density increased 143.7–156.25% for chopped plants and 140.9–161.0% for mushroom residues as compared with the control and measured at 180 and 360 days after the application of treatments. Other treatments did not show any significant effect at 180 days after their application. However, data measured at 365 days after treatment application showed that root density increased 34.2 and 40.9%, with the addition of pine chips and pine bark, respectively (*p* > 0.05). In contrast, the addition of rice husk did not show a significant effect on root density at any time (*p* > 0.05) (Figure 2A). Strikingly, the effect of those treatments that were statistically significant was only observed up to a depth of 30 cm into the soil profile (*p* < 0.05) (Figure 2A).

Microbe populations varied significantly after the addition of organic mulches. Chopped plants, mushroom residues, and pine bark increased populations of *Pseudomonas* sp. of the fluorescent group at 360 days after treatment application (*p* < 0.05). Cellulolytic microorganism numbers augmented at 180 days after application of chopped plants only. However, cellulolytic microorganism populations increased at 365 days after application of chopped plants, pine chips, and bark. There were more significant effect on *Trichoderma* sp. populations observed for any treatment applied (*p* < 0.05) (Figure 2B).

Microbial diversity (Shannon diversity index) increased 12.5 and 17.8% in chopped plants and mushroom residues treatments, respectively, at 180 days after application. In addition, at 365 days after the application of mushroom residues, pine chips and pine bark microbial diversity increased 15.6, 13.4, and 11.3%, respectively, for each treatment (Figure 3A). Microbial activity (respiration) augmented at 180 days after application of chopped plants (18.9%) and mushroom residues (23.4%) (*p* < 0.05) (Figure 3B).

We found a reduction of extreme values of soil moisture and soil temperature when using plant covers. Soil moisture within the first 20 cm of the soil profile and soil temperature showed lower values in the treatments with plant covers than the maximum soil temperature and moisture registered for the control. Similarly, soil moisture and temperature were higher in the treatments with plant cover than the minimum recorded for the control (Figure 4A,B). These findings mean that plant covers reduce the extreme maximum and minimum of soil temperature and moisture, thereby protecting avocado plants from stresses caused by events of heat, cold, drought, or waterlogging.

### 2.3. Evaluation of Agronomical Management Practices of Avocado Crops Aimed at Mitigating Adverse Effects of Climate Variability under the ENSO Phenomenon: El Niño and La Niña and Reduction of Incidence and Mortality of Diseases under Field Conditions

Significant differences of RRI and root density values were observed between treatments tested under the ENSO and standard climatic conditions (*p* < 0.05). During El Niño, root density of the native rootstock T3 significantly (*p* < 0.05) increased 38.0 and 87.5% at a depth of 0–30 and 30.1–50 cm, respectively as compared with the control. Values observed for T3 were followed by those observed for rootstock T1. No significant differences (*p* > 0.05) were observed between the control and rootstock T2. Meanwhile, during La Niña, rootstock T2 showed an outstanding increment in root density of 91.6 and 50% at a soil depth of 0–30 and 30.1–50 cm, respectively as compared with the control (*p* < 0.05). During standard climatic conditions, rootstock T1 showed the highest increase in root density of 68.7 and 58.3% at a soil depth of 0–30 and 30.1–50 cm, respectively as compared with the control (*p* < 0.05) (Figure 5A). Similarly, the variable RRI showed the highest values for treatments T3 during El Niño, T2 during La Niña, and T1 during standard climatic conditions (*p* < 0.05) (Figure 5B).

Disease incidence and severity values recorded in avocado trees were significantly different among the treatments applied for the oomycete *P. cinnamomi* and hypoxia, i.e., anoxia events. However, no differences were observed for the fungus *Verticillium* sp. (*p* > 0.05) (Figure 6A,B). During El Niño, the incidence and severity of the avocado wilt disease caused by *P. cinnamomi* decreased 150 and 97.2%, respectively, in T1 as compared with the control. T1 and T2 treatments exhibited better performance under La Niña conditions with a reduction of disease incidence of 52% for T1 and 64% for T2; and a reduction of severity of 68.8% for T1 and 52.3% for T2. Under standard climatic conditions, incidence and severity of the avocado wilt disease caused by *P*. *cinnamomi* decreased 50 and 40.1%, respectively, in T1 as compared with the control. In contrast, T2 and T3 showed a higher incidence and severity than treatment T1 (Figure 6A,B).

As expected, no effect was observed for events of hypoxia or anoxia during the drier conditions observed in El Niño. In treatment T2, a reduction of incidence and severity of 73.9 and 80%, respectively, was quantified under wetter La Niña conditions as compared with the control. Similar results were observed for T2 under standard climatic conditions with a reduction of disease incidence and severity of 53.1 and 63.3%, respectively as compared with the control (*p* < 0.05) (Figure 6A,B).

## 3. Discussion

### 3.1. Evaluation of Native Rootstock Adaptability to Different Soil Moisture Regimes, Tolerance to P. cinamomi and Verticillium sp., and Grafting Compatibility with Avocado cv. Hass under Net House Conditions

Native genotypes of avocado with morphological characteristics of the West Indian-, Mexican-, and Guatemalan-type races were studied for their adaptability to biotic and abiotic stresses. In the present work, we found a range of variability in the response of each native genotype to the edaphoclimatic factors evaluated. Native genotypes are usually the product of complex processes of hybridization among the three already mentioned races, which can be responsible, at least in part, for the great variability of responses observed in this work [19,20,43]. In general, West Indian-type genotypes showed better adaptability to higher soil moisture conditions and showed lower values of the avocado wilt disease caused by *P. cinnamomi*. These characteristics of high soil moisture correspond to some of the regions from where this race is native (i.e., tropical wet forest), which include vast regions of Colombia mainly located in the rainforests [21,40,41]. In addition, high soil moisture is highly conducive to the avocado wilt disease caused by *P*. *cinnamomi*. It is tempting to speculate that a native genotype could develop simultaneously tolerance to high soil moisture and to the avocado wilt disease as we observed in the present work, because both stresses were closely linked.

Guatemalan-type genotypes presented the highest grafting compatibility with avocado cv. Hass. These genotypes also showed advantages for drought conditions and intermediate tolerance to *P. cinnamomi*, *Verticillium* sp. and hypoxia/anoxia. Hass avocados are phylogenetically related to the Guatemalan-type race, which can explain the observed grafting compatibility with these genotypes [19,20,21,44]. The Mexican-type race exhibited the highest tolerance to *Verticillium* sp., a characteristic previously reported for this race [45,46]. Rootstock research looking for pathogen resistance especially for Phytophthora root rot, has been extensively developed in several regions around the world [40]. However, disease-tolerant rootstocks can be sensitive to other stress factors as has been observed for genotype Duke 7 that is mildly tolerant to *P*. *cinnamomi*, but sensitive to waterlogging. Advantages and disadvantages found, should encourage research for resistant genotypes with wide adaptability to diverse conditions present in avocado producing regions emphasizing the need of local research for the selection of the best rootstock genotypes for each different edaphoclimatic condition and aimed at the highest productivity.

Consistent with this premise, we used more than one single criterion combined for the selection of native avocado genotypes as resistance to the wilt complex caused by *P. cinnamomi*, *Verticillium* sp. and hypoxia/anoxia, performance under drought and flooding conditions for mitigation of climatic variability associated with the ENSO phenomenon [1,14] and graft compatibility. Here, we show four groups of genotypes clustered according to their responses to the stresses tested. The Guatemalan rootstock, with the best results for drought, graft compatibility, medium to low tolerance to pathogens, and hypoxia/anoxia, the West Indian rootstock group with high resistance to *P*. *cinnamomi* and hypoxia/anoxia, and medium tolerance to high soil moisture and drought and graft compatibility. Our results suggest that it is possible to find rootstocks with resistance and tolerance to combined stresses. However, as expected, resistance to all limiting stresses combined in one single genotype was not identified, indicating that research should continue with a large number of trials under different edaphoclimatic conditions. Such an approach could help in the identification of genotypes that could be useful as rootstock for local or regional edaphoclimatic conditions. This could be a more comprehensive and realistic approach to improve avocado crops in a tropical country with high edaphoclimatic variability such as Colombia and is in agreement with a precision agriculture-based approach [9,10,11,13].

### 3.2. Evaluation of Agronomical Management Practices to Mitigate the Adverse Effects of Climatic Variability Associated with the ENSO Phenomenon (El Niño and La Niña) and Reduction of Incidence and Mortality of Diseases under Field Conditions

Soil covers made of organic matter such as chopped plants and mushroom residues have proven to be useful for induction of root growth and development of avocado plants in the first 30 cm of the soil profile. Thus, they also promote better growth and development of avocado plants and higher tolerance to the avocado wilt disease caused by *P. cinnamomi* or *Verticillium* sp. Root system architecture traits have been extensively investigated as determinants of crop production in several crops including avocado [47]. In addition, different root traits have been identified as mechanisms of tolerance or resistance to different abiotic and biotic stresses [36]. For the AWC disease, several potential mechanisms of tolerance or resistance have been identified such as root regenerative ability, attractiveness to pathogen inocula, concentration and composition of root exudates, deposition of structural barriers such as callose and tylose, and the induction of molecular and biochemical defense pathways [48,49]. Similarly, Christie [50] found a direct correlation between AWC disease resistance and dry root mass, and confirmed that a faster regenerative capacity of the root system was an important factor in avocado genotypes to overcome the AWC disease, hence the importance of our findings on root induction and development.

In the present research, functional beneficial-type microbial groups, microbial activity, and microbial diversity increased with the addition of different sources of organic matter. Similar results have been reported to be beneficial for increasing plant health and nutrient availability (Appendix A) in a number of pathosystems and crops [37,51,52]. The addition of organic matter decreased the variability between the maximum and minimum values of soil temperature and moisture avoiding extremes, that could cause plant stress and possibly act as a buffer to better support periods of drought and high humidity, and improving tolerance to abiotic and biotic stresses [52]. The effects of the addition of organic matter to crops depends on a number of factors such as the source, composting degree, the addition of different amendments, edaphoclimatic conditions where the avocado orchard is grown, and many others. In addition, local availability and costs should be considered before including an organic matter source in an integrated crop management program. The analysis of research results suggests that local or regional research is mandatory to achieve successful results with the addition of organic matter in avocado crops.

Flat terrains are more prone to accumulate water into the soil profile that can create a favorable environment for AWC disease development. Nurseries can be an important source of inocula of several pathogens that are involved in the AWC disease. Therefore, production of seedlings with high quality and a better selection of the planting terrain by avoiding flat places that accumulate water in the soil profile and the sowing method (appropriate hole in the field), have been shown to be useful to reduce humidity accumulation into the soil profile, which, in turn, protects against disease development [18]. Integrating all the strategies analyzed in the present work resulted in better growth and development of avocado trees. It also reduced the incidence and severity of the avocado wilt complex disease and improved trees’ adaptability to the extreme climatic variation observed during the ENSO phenomenon. Rootstock genotypes should be selected considering several conditions such as adaptability to waterlogging and drought, resistance to main pathogens, graft compatibility, among others, because when they are selected taking into account only one of these conditions, they may not perform well with other conditions. The best results are observed when all crop management strategies available are combined to improve avocado production in the tropical highlands of Colombia.

## 4. Materials and Methods

### 4.1. Study Location

Data for this study were taken from avocado commercial lots for over an 8-year period (from 2009 to 2016). Experiments and evaluations were carried out in three lots planted with avocado cv. Hass grafted on 6-year-old West Indian rootstock, at a planting distance of 7 × 7 m. The first lot was located in the Municipality of Donmatias (6.496961 latitude and 75.412118 longitude, 2213 masl) in the Northern region of Antioquia, Colombia. The other two lots were located in the Municipalities of El Retiro (6.09715 latitude and 75.46478 longitude, 2100 masl) and La Ceja (5.95931 latitude and 75.41777 longitude, 2387 masl) in the Eastern region of Antioquia, Colombia. Details of edaphic conditions are shown in Appendix A. Laboratory and net-house procedures were carried out in the laboratory of Fitotecnia Tropical at Universidad Nacional de Colombia sede Medellín Núcleo El Volador (6°15′ N, 75°34′ W, 1496 masl).

The experiments were grouped into two phases. The first phase was divided into the following two parts: (i) selection of avocado native genotypes under net-house conditions (year 2009–2010) and (ii) evaluation of the effects of organic mulch application in commercial plots (year 2010–2011, plots 1, 2, and 3 described before). Phase two was performed using results obtained in the first phase. Three native avocado genotypes and the organic mulch with the best performance were selected to be evaluated together with agronomic management practices (described below in detail) aimed at improving adaptation of avocado trees to extreme climatic factors under field conditions (in Plots 1, 2 and 3, during years 2011 to 2016). For a better understanding of organization and schedule of experiments, Appendix A presents a scheme of each phase and experiment carried out. Details of all procedures and experiments are described below.

### 4.2. Phase One

#### 4.2.1. Evaluation of the Adaptability of Native Avocado Rootstocks to Different Soil Moisture Regimes, Resistance to *P. cinamomi* and *Verticillium* sp., and Grafting Compatibility with Avocado cv. Hass under Net House Conditions

Eleven native *P. americana* genotypes were collected from commercial orchards located in three avocado producing regions (i.e., North, East, and Southwest highlands) of the department of Antioquia, Colombia. The collected avocado native genotypes were identified according to the morphological characteristics of leaves, fruits, and seeds, in one of the Mexican-, Guatemalan- and West Indian-type races described in the literature) [19,20,21,43]. The avocado cv. Hass was used as a reference control, being the more widely exported variety in the world.

Seeds from native avocado genotypes were selected in the laboratory. Seeds weighing more than 20 g, obtained from apparently healthy fruits and harvested when they had 25–30% of dry matter, were selected. Subsequently, they were surface disinfested with 3% sodium hypochlorite for 3 min, followed by 3 min in sterile distilled water. As a pre-germination treatment, seeds were cut in the upper, lateral, and basal sides; then, they were sown for germination in autoclaved quartz (0.1 MPa, at 121 °C, for two cycles of 1 h each) and maintained at 50–70% of relative humidity. When seedlings exhibited 5 fully expanded leaves and the secondary root system showed good development as determined by visual inspection, cotyledons were removed to induce greater root formation. At that stage, seedlings were transplanted to 2 kg plastic pots with wet soil and placed under net-house conditions. The soil used for net-house experiments was an Andisol from the municipality of El Peñol, Antioquia Colombia. Details of the Andisol used are described in Appendix A. The soil was autoclaved (0.1 MPa, at 121 °C for two cycles of 1 h each). Plants were kept under net-house conditions, at 50% of the maximum soil moisture retention capacity for further use. In the soil in pots, we evaluated avocado plants under eight different soil moisture regimens (10, 20, 30, 40, 50, 70, 90, 110, 130, and 150% of the maximum soil moisture retention capacity). Monitoring of soil moisture was based on a calibration curve and quantified using values of volumetric humidity calculated with a V2 Moisture sensor (analogy, DF Robot™, reference SKU:SEN0114) [53].

To evaluate the adaptability of native rootstocks to conditions of excessive moisture or oxygen reduction (hypoxia/anoxia or waterlogging), we selected the values of maximum soil moisture retention capacity of 90, 110, 130, and 150%. In contrast, to evaluate the adaptability of native *P. americana* genotypes to drought, avocado plants were subjected to regimes of 10, 20, 30, and 40% of the maximum soil moisture retention capacity. To evaluate the resistance of avocado genotypes to the most important diseases *P. cinnamomi* strains (Code: PCSOC1, PC10, and PC8) and *Verticillium* sp. strains (Code: VAC3, VAC2, VACC4) were used. These pathogens were obtained from the collection of avocado pathogenic strains kept at the laboratory of Fitotecnia Tropical located at the Universidad Nacional de Colombia sede Medellín, which have been morphologically and molecularly characterized. The inoculum was grown in potato-dextrose agar medium (PDA-Difco, Detroit, MI, USA) at 22 °C for 10 days. Inoculation of avocado native genotypes evaluated as potential rootstocks was performed by adding in individual plants of an aqueous suspension containing inoculum of *Verticillium* sp. to a final concentration of 1 × 10^5^ mL^−1^ infective propagules and inoculum of *P*. *cinnamomi* at a concentration of 1 × 10^3^ mL^−1^ infective propagules at four equidistant points in each pot containing an individual avocado genotype [30]. 

Plant performance was evaluated using a specific scale designed for avocado stress associated with waterlogging, and *P*. *cinnamomi* and *Verticillium* sp. diseases [54], by measuring values of this scale every 10 days during a period of 120 days. Obtained data were used to estimate the area under disease progress curve (AUDPC) [55]. In addition, in the end of assay (120 days) total plant dry weight was determined by drying plant tissues at 60 °C in an oven (Binder^®^), until a constant weight was obtained. With dry weight values, we developed a mathematical model of relative rate of increase (RRI), using Equation (1) [56]. The response variable measured for this phase of experiments was plant dry weight increase through time represented by the RRI value.
(1)$$RRI=\frac{d\left(ln\left(w\right)\right)}{dt}$$
where *RRI* is the relative rate of plant weight increase, *d/dt* means temporal derivate, *ln* is a natural logarithm, *w* is weight, and *t* is time.

To evaluate graft compatibility, selected avocado genotypes were used as rootstocks and the Hass variety as scion. Plants which were three to five month sold, or taller than 30 cm, of each of the native avocado genotypes selected were used as rootstocks for graft compatibility experiments. Plant dry biomass was evaluated every 10 days for 180 days using the methodology described above. In addition, the presence of malformations and anomalies in the grafted area was recorded.

For all three experiments, a completely random experimental design was used with five replicates per treatment. The experimental unit consisted of 15 seedlings and the evaluation had two repetitions through time (year 2009—2010). In each variable (RRI and AUDPC), data homoscedasticity and normality were verified using criteria of Levene and Kolmogorov–Smirnov, respectively. Subsequently, data were subjected to analysis of variance and means were compared by the Tukey test with a significance of 95% (*p* ≤ 0.05). In addition, a principal component analysis was performed and responses of different genotypes to soil moisture, adaptability to hypoxia/anoxia, disease development by *P. cinnamomi* and *Verticillium* sp., and grafting compatibility were grouped by cluster analysis using the k-means clustering method. The analyses were performed using the R computing software [57].

#### 4.2.2. Effect of Organic Mulch Addition on Avocados Roots Growth, Microbial Populations, and Reduction of Stress Associated with Environmental Conditions under Field Conditions

Experiments in each of the three lots (Plot 1, 2 and 3, described before) planted with avocado cv. Hass grafted on 6-year-old West Indian rootstock were performed in a random block design (each lot) with five replicates and ten plants as an experimental unit. The experiment was developed during 2010–2011. Treatments consisted of the addition of different organic mulch as follows: (i) pieces of chopped plants of less than 2 cm mixed in equal proportions (*Pennisetum clandestinum/Cynodon plectostachium/P. americana, Pteridium aquilinum,* and *Acacia melanoxylon*); (ii) mushroom residues + peat (1:1); (iii) chips of pine (*Cupressus lusitanica* and *Pinus patula*) of less than 5 cm in size ((1:1) cut residues of industrial processing); (iv) rice husk previously washed with water for cleaning and elimination of herbicide residues and (v) pieces of pine bark (*C. lusitanica* and *P. patula*) of less than 1 cm. During this work crop management practices were performed following the farmer directions without additional intervention from researchers.

Each mulch was added to avocado plants in a 15 cm thick layer applied over the entire drip of the tree, in a diameter of 2 meters around the base of the stem. In addition, the average decomposition time of each mulch was determined. For this, 20 grams of each mulch were oven-dried at 65 °C. In each experimental unit, 9 bags of mulch were spread (15 cm diameter × 30 cm high) over a plastic mesh of 2 mm pore size [58]. Analysis was performed during three time periods (beginning (Day 0), intermediate (Day 180) and final (Day 365)). At each time period a quantity of mulch corresponding to three bags was removed and dry biomass quantified in the laboratory. As complement, viable roots were quantified from each plant by destructive sampling. A stainless-steel cylinder (100 cm^3^ volume) was inserted in 10 points distributed around a diameter of 2 m from the base of the plant stem at a depth of 1 m. The removed material was washed with tap water selecting avocado roots and determining their status as non-viable diseased roots (necrosed) and viable healthy roots (white) [30]. Remaining roots and rhizosphere soil samples were kept for quantification of microorganisms in the laboratory.

Horizontal and vertical root dry biomass were quantified using data of viable roots. Root samples were dried at 60 °C in an oven (Binder^®^) until a constant weight was obtained. Root density was quantified as the ratio between total biomass and the volume occupied by roots using Equation (2). In Equation (2) it was assumed that the root exploration volume (horizontal and vertical growth) presented a cylindrical-like geometry.
(2)$$V=\pi r^{2}h$$
where *V* is the cylinder-like volume occupied by roots, *r* is the cylinder radius (horizontal root growth), and *h* is the cylinder height (vertical root growth).

Populations of microbial functional groups, previously identified as beneficial in avocado crops (i.e., *Pseudomonas* spp., *Trichoderma* spp., and cellulolytic microbes), and ecological variables of microbiota (activity and biodiversity) were measured in each experimental unit during the three stages of the experimental process (i.e., beginning (Day 0), intermediate (Day 180) and final (Day 365)) [18,59].

For quantification of microbe functional groups and microbial activity and biodiversity, 5 g of roots and rhizosphere soil were macerated in 200 mL of sterile distilled water, and serial dilutions (1 × 10–1^−1^ × 10^−5^) were prepared in sterile distilled water (*p:v*). *Pseudomonas* spp. colonies were grown in commercial Pseudomonas (Merck^®^) culture medium and fluorescent pigments were identified under ultraviolet light (260 nm) [60]. *Trichoderma* spp. colonies were identified by morphology and quantified in PDA culture medium supplemented with 0.5% Igepal (Sigma, St Louis, MO, USA) [18,61]. For quantification of cellulolytic microorganisms, we followed the methodology reported by Wood [62]. Microorganisms were quantified as colony forming units (CFU) by direct counting from the serial dilutions performed.

Cultivable bacteria, fungi, and actinomycetes were considered for the microbial diversity analysis. Bacteria were evaluated on nutrient agar (NA) at pH 6 for 48 h at 28 °C with the addition of benomyl (Benlate^®^) (75 mg L^−1^). Fungi were evaluated in potato dextrose agar medium (PDA) at pH 5.0 adjusted with lactic acid and supplemented with streptomycin sulfate (100 mg L^−1^) and chloramphenicol (30 mg L^−1^). Fungal colonies were incubated for 6 days at 25 °C. Actinomycetes were quantified in starch-casein (SC) medium supplemented with 0.25% fluconazole incubated at 28 °C for a period of 10 days. Different morphotypes were identified using morphological characteristics of microorganisms recovered in the different media culture. The Shannon diversity index was calculated using the PAST program version 2.16 using data of different morphotypes [63]. Total microbial activity was measured by the respirometry technique [64].

For each dataset (decomposition, root density, CFU, microbial activity, and microbial diversity) homoscedasticity and normality were verified using the criteria of Levene and Kolmogorov–Smirnov, respectively. In order to comply with these principles, inocula data were logarithmically (ln) transformed and subjected to analysis of variance. Then, means were compared by the Tukey test with a significance of 95% (*p* ≤ 0.05).

### 4.3. Phase Two

#### Agronomical Practices in Avocado to Mitigating Adverse Effects of Climate Variability under the ENSO Phenomenon and Incidence and Mortality of Diseases under Field Conditions

In this section, we describe the evaluation of four treatments in avocado fields. Treatments were designed using the information obtained in preliminary experiments under net-house and field conditions and previous results from different studies conducted under field conditions [28,30,31,37]. Such results included parameters associated with production of seedlings in the nursery, planting techniques, and the use of organic mulch. Standard parameters used by farmers for fertilization and pest, weed, and disease management were followed, according to the criteria of the technical assistants. The experiment was developed under field condition (Plots 1, 2, and 3 described before). During this work, crop management practices different from the treatments evaluated were performed according to specific indications of farmers without additional intervention of researchers (e.g., planting distances, fertilization, pest and disease management, pruning, drainage, among others).

The traditional system implemented in avocado producing farms was used as the control. Usually, farmers buy avocado plants of the Hass variety grafted on uncharacterized Indian West rootstock in commercial nurseries. Plants are usually 12 months old and grown in a plastic bag of 43 cm long and 20 cm in diameter. Sowing in the field was performed in holes of 60 cm depth and 60 cm of diameter on average and no mulch was added. Treatments implemented in the present research were: T1: A new proposal, consisted of the genotype selected that showed the best performance in the aforementioned tests developed to determined resistance and tolerance to biotic and abiotic stresses (Table 1 and Figure 1). The selected genotype corresponded to GX3 (of Guatemalan-type race). The entire process of seedling production was carried out within the farms. Seed disinfestation and pre-germination treatments were performed as described above [31]. The seed germination process, bag size selection, growing substrate, grafting, length of time at the nursery stage, and phytosanitary measures were carried out as previously reported [30,31]. Seedlings were transferred from nursery to field conditions where they were planted in holes of 30 cm depth and 60 cm in diameter and in mounds. Just at the moment of planting a layer of 15 cm thick of mulch (chopped plants and mushrooms, proportion 1:1) was added in 60 cm around the base of the stem. T2: The same practices as in T1, but the native avocado genotype AX4 was used instead. The AX4 avocado genotype showed the best performance under high soil moisture and hypoxia/anoxia conditions (tolerance to waterlogging) (Table 1). T3: The same practices as in T1, but the native avocado genotype GX2 that performed better under conditions of low soil moisture (tolerance to drought) was used instead genotype GX3 used in T1 (Table 1).

A random block (each lot) was used as experimental design with five replicates and 15 plants as an experimental unit. RRI and root density were quantified every two months, as described above. In addition, disease incidence and mortality of avocado trees caused by the three major causal agents of AWC (i.e., *P. cinnamomi, Verticillium* sp., and hypoxia/anoxia) were quantified as described [37].

The experiment in avocado plots was carried out for 6 years (from 2011 to 2016), during which extreme climatic conditions associated with the ENSO phenomenon (La Niña, El Niño) took place [18,65]. The following three periods when the ENSO phenomenon happened during the experimental time period were selected for further analysis: La Niña (July to December 2011), standard conditions (July to December 2013), and El Niño (July to December 2015). Each period was about 6 months in duration (Figure 7). Climate variables were measured using a low-cost electronic device containing an automatic cube type pluviometer (Argent Data Systems) for precipitation, a DHT11 sensor (digital, D-Robotics) for humidity and temperature, a V2 Moisture sensor (analogy, DF Robot, reference SKU:SEN0114) for soil moisture, and a DS18B20 sensor (digital communication, DF Robotin) for soil temperature [53]. The analyses of change (increases and decreases) in temperature and precipitation under ENSO phenomenon (El Niño, La Niña) were carried out analyzing the variation of these variables in the experimental plots with respect to historical series (2070–2010) of data obtained from nearby IDEAM weather stations. The variables described above (i.e., root density, RRI, and incidence and mortality caused by AWC) were evaluated every two months during the three periods of evaluation described.

For statistical analysis, a mixed univariate analysis of variance (ANOVA) was used assuming plants as the fixed within-subject factor, each of the events and the three periods of evaluation selected as the random factor and variables as a response. Treatments were independently evaluated during each phase of the phenomenon (El Niño, La Niña, and standard conditions). Data homoscedasticity and normality were evaluated using the Levene and Kolmogorov–Smirnov tests, respectively. Since the number of plants in each category was different due to natural mortality, the tests proposed by Dunnett’s and Tukey–Kramer for multiple comparisons of samples with different sizes, was applied. For test calculations, an algorithm implemented in the DTK package available in the software R was used [66].

Procedures to diagnose diseases and disorders associated with the avocado wilt complex disease in experimental plots included symptom identification and microorganism isolation in semi-selective media followed by morphological characterization. For biotic causal agents, microbe identification was confirmed by sequence analysis of the genomic ITS regions. For abiotic causal agents such as hypoxia/anoxia and root atrophy, aerial symptoms, root dissection, and verification of microorganism presence and absence were performed [30].

## 5. Conclusions 

Within the *P. americana* genetic pool, we identified genotypes that could be useful as rootstocks for avocado cv. Hass crops. These genotypes proved to be adaptable to extreme conditions of soil drought and flooding and to infection by the microbial pathogens *P*. *cinnamomi* and *Verticillium* sp. Using several combined strategies such as organic mulch addition (chopped plants and mushrooms), rootstock selection (Guatemalan-type genotype), and new tools for planting (plants with genetic identity, free of pathogen and root damage, and planted under conditions of low alteration of the micro-drainage networks), it was possible to mitigate the adverse effects on avocado crops caused by extreme events of climatic variability such as those observed during the ENSO phenomenon.

## Figures and Tables

**Figure 1 plants-09-00790-f001:**
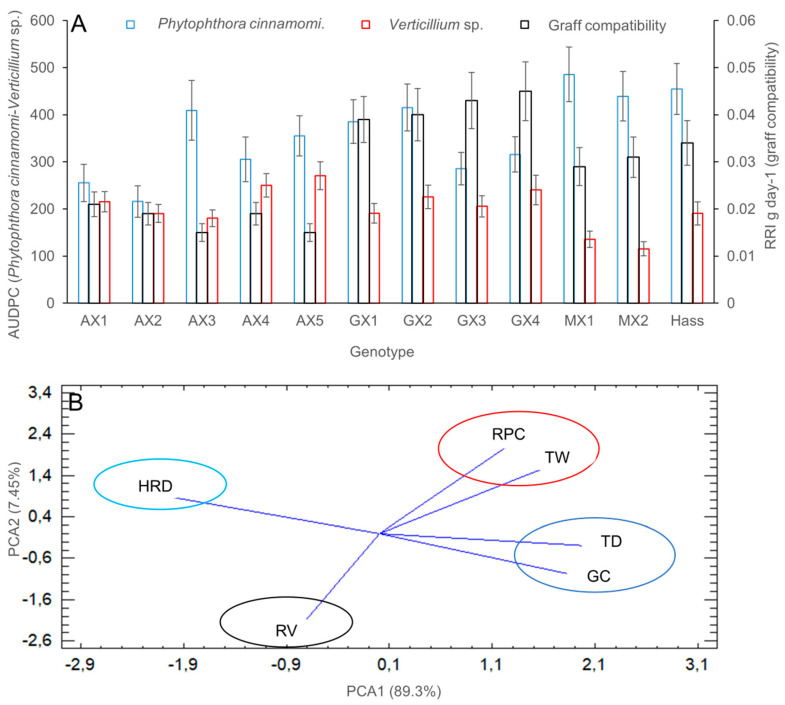
Tolerance of native rootstocks of *P. americana* to *P. cinnamomi* and *Verticillium* sp., and graft compatibility with cv. Hass. (**A**) AXN, morphotype of *Persea americana* var. *americana* (West Indian-type race); GXN, morphotype of *Persea americana* var. *guatemalensis* (Guatemalan-type race); MXN, morphotype of *Persea americana* var. *drymifolia* (Mexican-type race); RRI, relative rate increase (g day^−1^); AUDPC, area under disease progress curve; error bars represent the confidence intervals of the mean, validated by the Tukey mean separation test and no overlapping of the error bars indicates significant differences (*p* < 0.05); (**B**) Clustering of genotypes based on evaluated agronomic variables. RPC, resistance to *P. cinnamomic*; TW, tolerance to waterlogging; HRD, high rate of developed under extreme soil moisture; TD, tolerance to drought; GC, graft compatibility with cv. Hass genotype; RV, resistance to *Verticillium* sp.

**Figure 2 plants-09-00790-f002:**
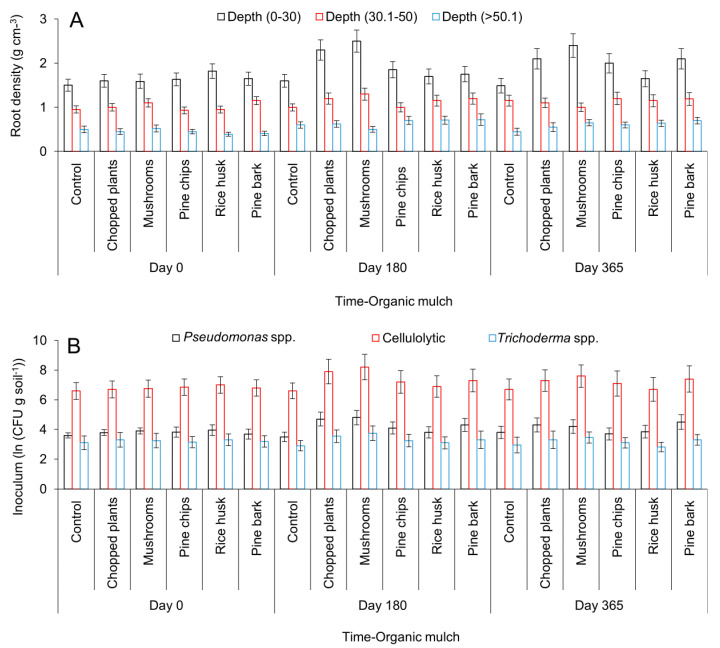
Root densities (**A**) and rhizosphere microorganism populations (**B**) measured after the addition of organic mulch in avocado plants. Error bars represent the confidence interval of the mean, validated by the Tukey mean separation test. No overlapping of the error bars indicates significant differences (*p* > 0.05).

**Figure 3 plants-09-00790-f003:**
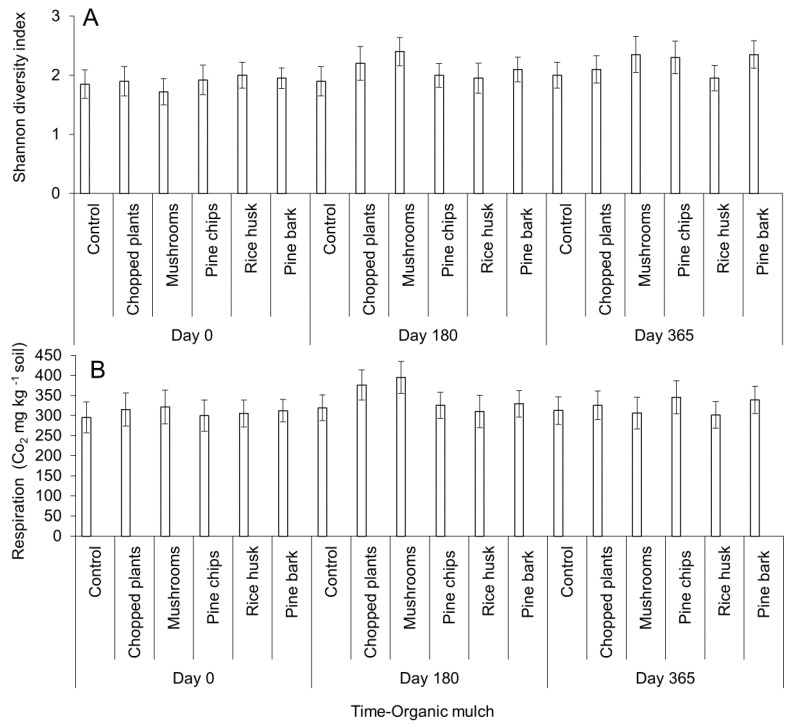
Microbial diversity (**A**) and microbial activity (**B**) values after the addition of organic mulch in avocado plants. Microbial diversity was calculated using the Shannon index. Microbial activity is proportional and represented as respiration values. Error bars represent the confidence intervals of the mean, validated by the Tukey mean separation test. No overlapping of the error bars indicates significant differences (*p* > 0.05).

**Figure 4 plants-09-00790-f004:**
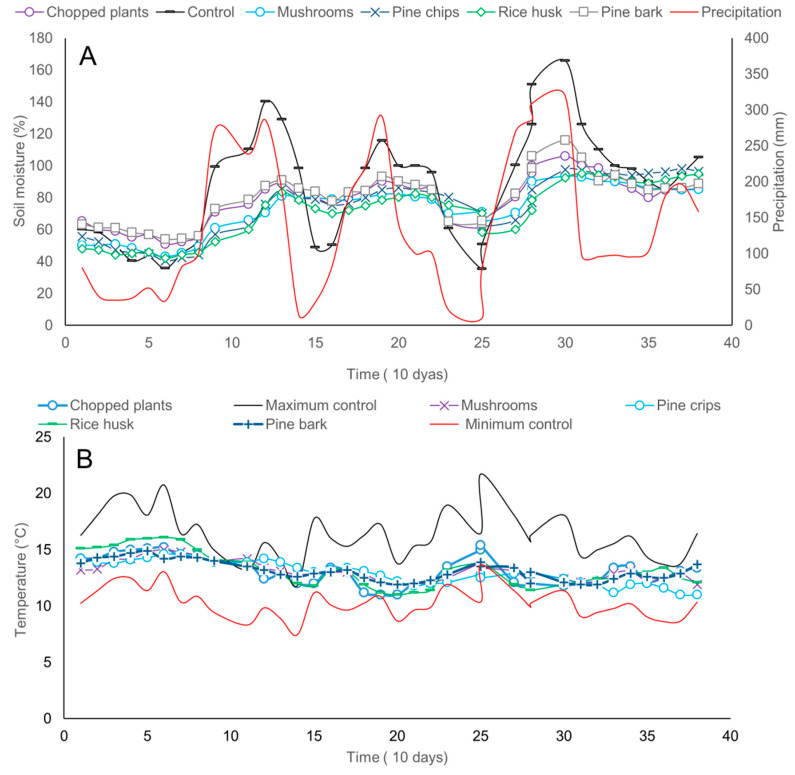
Dynamics of soil moisture (**A**) and temperature (**B**) within the soil profile through time as a result of the addition of organic mulch in avocado plants.

**Figure 5 plants-09-00790-f005:**
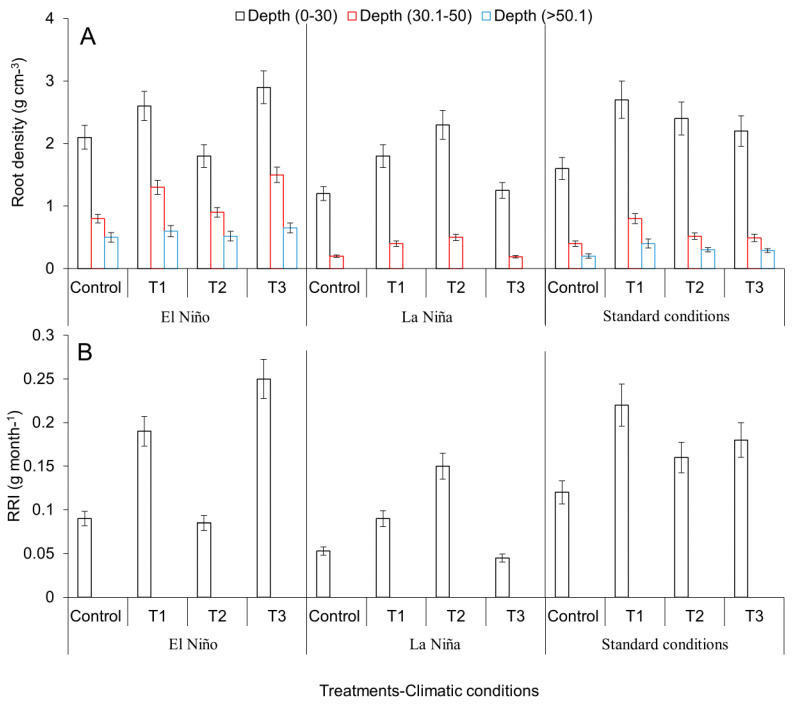
Variation of root density (**A**) and relative rate increase of dry plant weight (**B**) values as effect of the application of agronomical practices aimed to mitigate the adverse effects of the ENSO phenomenon (El Niño and La Niña) under field conditions. T1, new tools (plants with genetic identity, free of pathogen and root damage and planted under conditions of low alteration of the micro-drainage networks) implemented under field conditions and the native avocado genotype with the best performance to waterlogging, drought, graft compatibility and resistance/tolerance to *P. cinnamomi* and *Verticillium* sp; T2, practices implemented in T1 + the native avocado genotype with the best performance for waterlogging conditions; T3, practices implemented in T1 + the native avocado genotype with the best performance under drought conditions; Control, traditional system implemented in avocado producing farms. Error bars represent the confidence intervals of the mean, validated by Dunnett’s, and Tukey–Kramer separation tests. No overlapping of the error bars indicates significant differences (*p* > 0.05).

**Figure 6 plants-09-00790-f006:**
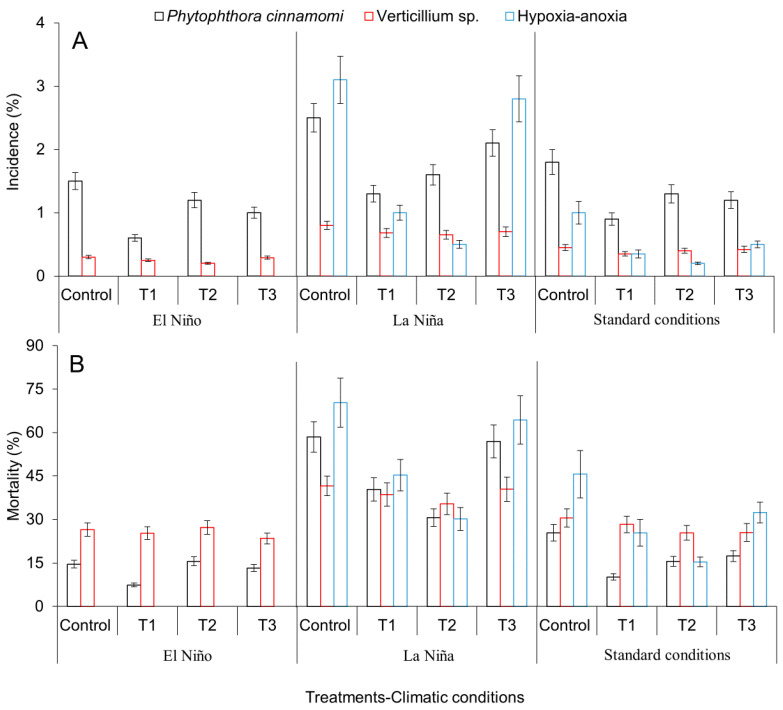
Variation in incidence (**A**) and mortality (**B**) values of avocado wilt disease induced by major causal agents after agronomical practices aimed to mitigate the adverse effects of the ENSO phenomenon (El Niño and La Niña) under field conditions. T1, new tools (plants with genetic identity, free of pathogen and root damage and planted under conditions of low alteration of the micro-drainage networks) implemented under field condition and the native genotype with the best performance to waterlogging, drought, graft compatibility, and resistance to *P. cinnamomi* and *Verticillium* sp.; T2, practices implemented in T1 + the native genotype with the best performance under waterlogging conditions; T3, practices implemented in T1 + the native genotype with the best performance under drought conditions; Control, traditional system implemented in avocado producing farms. Error bars represent the confidence interval of the mean, validated by Dunnett’s, and Tukey–Kramer separation tests. No overlapping of the error bars indicates significant differences (*p* > 0.05).

**Figure 7 plants-09-00790-f007:**
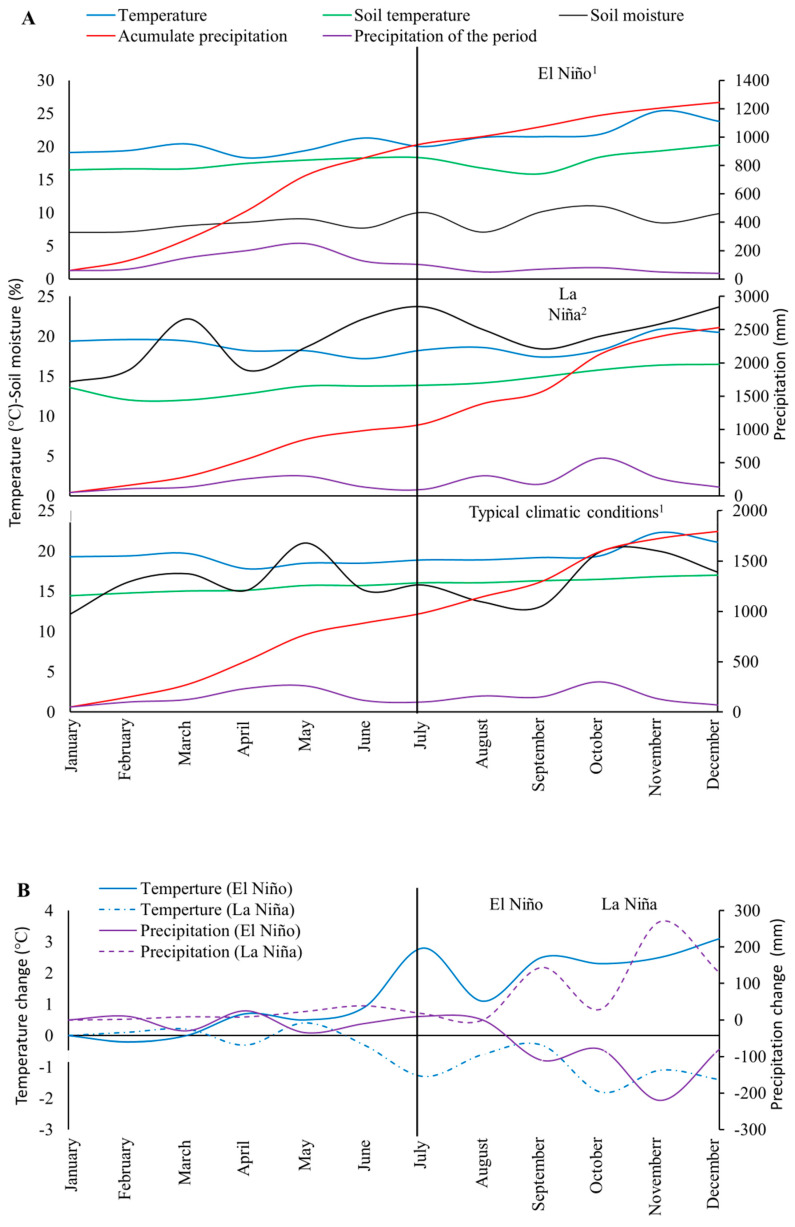
Climatic conditions under ENSO phenomenon (El Niño-La Niña) and typical standard conditions in the regions where experimental plots of avocado crops were located. (**A**) Temporal variation of climatic variables in the lots evaluated. ^1^ Soil moisture (real value/5) and ^2^ Soil moisture (real value/7); (**B**) Change in climatic variables under ENSO phenomenon.

**Table 1 plants-09-00790-t001:** Adaptability of native avocado rootstocks to different soil moisture regimens.

Genotype/Soil Moisture in Percentage (%)	10	20	30	40	50	70	90		110		130		150	
	RRI ^1^	RRI ^1^	RRI ^1^	RRI ^1^	RRI ^1^	RRI ^1^	RRI ^1^	AUDPC ^2^	RRI ^1^	AUDPC ^2^	RRI ^1^	AUDPC ^2^	RRI ^1^	AUDPC ^2^
AX1	0.004 c	0.008 c	0.015 c	0.023 c	0.047 c	0.032 c	0.059 a	12.5 c	0.04 b	35.4 e	0.025 a	95.4 c	0.006 a	200.3 c
AX2	0.003 d	0.009 c	0.016 bc	0.024 c	0.050 b	0.034 c	0.05 b	10.1 c	0.035 b	30.2 e	0.021 a	88.9 c	0.007 a	185.4 c
AX3	0.005 c	0.010 c	0.017 b	0.025 c	0.055 b	0.038 b	0.065 a	19.3 bc	0.052 a	60.3 d	0.018 ab	150.4 b	0.005 a	230.4 bc
AX4	0.005 c	0.011 c	0.019 b	0.026 bc	0.059 ab	0.04 b	0.069 a	7.5.6 c	0.049 a	25.8 de	0.02 ab	70.3 c	0.006 a	165.6 bc
AX5	0.002 d	0.008 c	0.014 c	0.022 c	0.045 c	0.031 c	0.045 c	13.8 c	0.038 b	50.3 d	0.015 b	100.3 c	0.006 a	205.8 c
GX1	0.006 b	0.013 b	0.024 a	0.031 a	0.065 a	0.047 ab	0.050 b	25.6 b	0.028 c	75.4 c	0.014 b	160.3 b	0.001 b	225 bc
GX2	0.012 a	0.025 a	0.029 a	0.032 a	0.066 a	0.045 b	0.045 c	28.4 b	0.025 c	89.3 bc	0.009 c	185.4 ab	0.001 b	250.3 b
GX3	0.010 a	0.020 a	0.027 a	0.034 a	0.070 a	0.052 a	0.055 b	20.3 bc	0.03 c	80.2 c	0.008 c	160 b	0.001 b	230.4 bc
GX4	0.007 b	0.013 b	0.026 a	0.031 a	0.061 ab	0.045 b	0.047 c	32.4 b	0.023 d	100.5 b	0.011 c	205.3 a	0.0009 b	301.2 a
MX1	0.008 a	0.014 b	0.020 b	0.023 c	0.042 c	0.025 d	0.032 d	39.1 a	0.028 c	115.8 a	0.007 c	215.6 a	0.0008 b	320.5 a
MX2	0.006 b	0.015 b	0.018 b	0.026 bc	0.045 c	0.027 d	0.035 d	42.3 a	0.024 cd	120.3 a	0.006 c	225.4 a	0.006 b	335.1 a
Hass	0.007 b	0.013 b	0.017 b	0.029 b	0.052 b	0.035 c	0.038 d	32.1 b	0.027 c	100 b	0.007 c	190.3 a	0.0009 b	250.2 b

AXN, morphotype of *Persea americana* var. *americana* (West Indian-type race); GXN, morphotype of *Persea americana* var. *guatemalensis* (Guatemalan-type race); MXN, morphotype of *Persea americana* var. *drymifolia* (Mexican-type race);^1^ relative rate increase of dry plant weight (g day^−1^); ^2^ area under the disease progress curve. Different letters mean significant differences (*p* < 0.05) based on the Tukey mean separation test.

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
