# Peer review of "Mitigation of the Adverse Effects of the El Niño (El Niño, La Niña) Southern Oscillation (ENSO) Phenomenon and the Most Important Diseases in Avocado cv. Hass Crops"

_plants, 2020, doi:10.3390/plants9060790_

Round 1
Reviewer 1 Report
I don't not seen authors' responses to each of my previous comments. I can not review and evaluate without reviewing authors' responses on how they are addressed. I am returning the manuscript.
Author Response
This letter serves to re-submit a manuscript with code“787935”, by myself and my colleagues Juan Camilo Henao Rojas and Juan Gonzalo Morales Osorio, for consideration for publication in the important journal Plants. We found the reviews to be helpful, and we have responded to each of their comments in detail. Based on the reviewer’s comments, we have made modifications to the original manuscript and proofread it carefully.
We detail our responses to each of the comments below, and we believe that the manuscript has been highly improved. We hope that this contribution will now prove acceptable for publication in your journal. We refer to line numbers. In addition, the language of the document thoroughly reviewed and corrected as requested, process made by an expert in English language.
Response
Reviewer 1
General comments
- This study was very difficult to comprehend. I think that the authors surveyed different areas and also conducted an experiment. I feel their aim is to understand if management practices may help to moderate (lessen) disease expression that tends to increase with specific weather events
R: The work did not consist of surveys, the process was carried out under a strict process of scientific method. A design, experimental units and clear repetitions were used. In addition, a rigorous statistical analysis was carried out. Please see material and methods lines: 419-428 (the first experiment); 431-441 and 486-490 (the second experiments); 503-520, and 521-539 (the third experiment). In addition, we presented a scheme of experimental phases developed under net house and field conditions. See lines 594-596.
- I have barely made it through the manuscript (pg 2 of 20) and already offered 20 suggestions. This is a clear indication that this manuscript is not ready to be reviewed and needs to be returned to the authors to have the writing improved. Once the writing is improved, reviewers can then critique the science.
R: The language of the document thoroughly reviewed and corrected as requested, process made by an expert in English language. In addition, the document was reviewed careful and the quality was improve.
Specific comments
- L2-3 “ENSO” is vague since each region is affected differently. Don’t assume readers will understand your title’s context. Can you be more specific as to the conditions (e.g. cool & wet) that increase disease expression?
R: We added more details. Please see lines 2-6.
- L18 “Among the major limitations” What limitations? You need a better transition between sentence one and two of abstract. L18 “ biotic or abiotic” Disease expression is typically a combination of biotic and abiotic conditions not one or the other. L20 “ the adverse effects associated with the ENSO phenomenon (El Niño-La Niña) and the most important diseases” Still not really clear what you’re talking about. I think you are wanting to say something like understand how to moderate disease outbreaks in response to ENSO events. L22 You list (some) experimental treatments but don’t mention your response variable(s). Also, the next sentence indicates you varied soil moisture. But this is not clear from L22. Writing has a lot of information gaps. L25 “ with several grafts of compatibility with the cv” This does not make sense. What is “cv” (coefficient of variation)? L26 “coverages” Huh? L26-8 “ adequate production of seedlings and planting field decrease the susceptibility to adverse effects associated with the ENSO phenomenon (El Niño-La Niña) and the incidence of diseases under climate variability” Does not make sense.
R: We rephrased this part and added more details of the abstract to make it easier to understand. Please see lines 18-37.
- L33 “ commonly planted commercial variety [of what?] in the internal market [what?] and for export in the world.” Wordy and confusing. L39-40 Unclear & confusing. L41 “or” (?).L43 Again, writing is wordy and confusing.
R: We rephrased this part and added more details of the abstract to make it easier to understand. Please see lines 63-68 and 69-81.
- L45 This content is more general than the prior paragraphs. Don’t you think you should start your introduction with the most general content and move toward specifics? You need to reorganize the content to improve flow. Also, it seems you should be focusing from the start on how crop disease(s) is impacted by climate. Not sure if climate change needs to be included or not since you seem to be focusing on ENSO. Either way, pick one and keep it is as the focus throughout.
R: We agree with the reviewer and moved the order. In addition, we rephrased this part and added more details. Please see lines 40-48.
- L46 “CO2” is directly related to disease? L48 “These potential impacts may be more visible in tropical regions” vague. L49 “Colombia is a country that naturally presents a high climate variability”- wordy. L50 “territory” but previous line you call it a country. L51 What changes? Vague and confusing. L56 what conditions? Again, the writing is very vague and lacks useful descriptions to help readers understand what is going on.
R: We reworded these sentences for clarity. Please see lines 40-48; 49-56; 57-62.
- L231 Was this an experiment that follows a common experimental design or a survey of fields with different properties?
R: This paragraph only refers to the location of the lots, not to the experimental design. In addition, a rigorous statistical analysis was carried out. Please see material and methods lines: 419-428 (the first experiment); 431-441 and 486-490 (the second experiments); 503-520, and 521-539 (the third experiment). In addition, we presented a scheme of the experimental phases developed under net house and field conditions. See lines 594-596.
- L234 7x7 is an area not a distance
R: refers to the planting distance in avocado crops and this form is commonly used in the literature.
- L317 “The treatments 317 consisted on the addition of different organic mulch:” wordy and confusing
R: We reworded these sentences for clarity. Please see lines 431-441.
- L378 Can you give a succinct description of the treatments here?
R: We reworded these sentences for clarity. Please see lines 493-520.
Reviewer 2
General comments
- Introduction: This section is very weak. Needs more details on defining the problem and previous work done related to this. No explanation of ENSO related literature review on either avocado or other comparable cropping systems was given. I encourage authors to improve Introduction overall and add more relevant literature reviews.
R: We agree with reviewer and in the introduction we included a better description of all aspects as suggested. Please see lines 49-56; 57-62.
- Materials and Methods: Not sure why results are presented before methodology in the manuscript (a). I have not seen T1,T2, and T3 clearly defined in methods (b). Also, how did you classify ENSO years. And why only 1 year of El Nino, La Nina and Neutral was considered (c). If historical data available, this study needs to consider ALL available El Nino, La Nina, and Neutral years to make results (d) and conclusions more robust (e). Eq.2 is so basic and not necessary (f).
R:
- This order follows the format of the journal.
- We reworded these sentences for clarity. Please see lines 493-520
- We rephrased this part and added more details to make it easier to understand. Please see lines 526-539 and 547-554.
- We agree with reviewer and in the material and methods we included the Figure 7. Please see lines 547-554.
- We rephrased this part and added more details to make it easier to understand. Please see lines 529-566.
- This equation is important since it was used to explain a specific phenomenon, not common like root architecture.
- Results: Authors mentioned that during El Niño, root density in T3 (native rootstock) significantly (P<0.05) increased 38.0 and 87.5% at a depth of 0-30 and 30.1-50 cm, respectively, compared with the control; In contrast, during La Niña, T2 showed an outstanding increment in root density of 91.6 and 50% at a soil depth of 0-30 and 30.1-50 cm, respectively, when compared with the control (P<0.05).This is interesting results however; authors MUST explain the RATIONALE for these significant differences between El Nino, La Nina, and neutral years. Same for RRI, Incidences, and Mortality (a). I suggest addition another subsection in discussion to provide justification and reasoning for these statistically significant differences during ENSO events (b). Authors should also consider adding chart or table summarizing weather variables, temperature and precipitations, during El Nino, La Nina, and neutral years considered in this research (c).
- We rephrased this part and added more details to make it easier to understand. Please see lines 281-287 and 303-315.
- We appreciate the reviewer concerns, and indeed this contribution represents something superficial of the usual context for this topic. Nonetheless, the contribution is relevant to the broader topic we are not physiologists or climatologist and therefore cannot speak more directly to that topic, but we offer the contribution as a view from a distinct toolset that we hope will offer some insights into a complex topic that has not been treated for this crop to date.
- We agree with reviewer and the material and methods we included the Figure 7. Please see lines 547-554.
- I didn’t find conclusion section.
R: We included this part and added more details to make it easier to understand. Please see lines 547-554.
- Miscellaneous: There are many TYPOS that needs to be cleaned up.
R: The language of the document thoroughly reviewed and corrected as requested, process made by an expert in English language. In addition, the document was reviewed careful and the quality was improve.
Round 2
Reviewer 1
- I don't not seen authors' responses to each of my previous comments. I can not review and evaluate without reviewing authors' responses on how they are addressed. I am returning the manuscript.
R: All corrections were incorporated in the re-submitted document, and each consideration was considered in the response letter described in the previous paragraphs.
Reviewer 2
General comments
- The subject of the manuscript is a hot topic under the context of the prevailing and forecasted climatic conditions. In fact, urge the necessity to concentrate efforts in the search for strategies that mitigates the climate change and variability effects on agricultural productivity. The introduction section explains well the motivation of the study exposing a clear big objective, still, it is missing information about the plant responses evaluated on the experiment, why are they important to be monitored in order to meet the proposed objectives. The introduction also fails by do not explain how the avocado wilt complex (AWC), the object of study in this manuscript, affects the plants (a). Moreover, no introduction is made to the proposed adaptation practices, specially the use of organic mulches (b) .
- We agree with reviewer and information about AWC was added. Please see lines 75-86.
- We agree with reviewer and information about proposed adaptation practices were added. Please see lines 87-105.
- Material and methods section should be improved, especially on the experimental design and conditions description, which are a little confuse. It is a big study, with different experiments and a lot of variables incorporated, which may have been performed in different chronological times and in different geographical locations, and with specific objectives. The reader ends up getting lost, being necessary a huge effort to understand the entire experiment. I suggest to better organize the experimental design description, for instance by an elucidative schematic representation, incorporating the different experiments and the respective conditions, the relationship between each other, and the methodologies used in each experiment to monitor the plant responses.
R: We included this part and added more details to make it easier to understand. Please see lines 346-352-. In addition, we presented a scheme of the experimental phases developed under net house and field conditions. See lines 594-596.
- In the discussion section, the performance of the different selected genotypes described in 2.3, i. e. under the ENSO phenomenon, was less discussed. The conclusions are somewhat vague.
R: We included this part and added more details to make it easier to understand. Please see lines 281-287 and 547-554.
Specific comments
Regarding the formatting issues please verify carefully, I just point some examples:
- Line 17: “have had”
R: done
- Line 85: Grafting capital letter
R: done
- Line 97: TRI or RRI?
R: done
- Figure 1 and respective caption: the acronyms did not fit. TPC and THA or RPC and TW? TV or RV?
- Is after the respective description, but (B) is before, please uniformize.
R: done
- Line 149: “ofe”
R: done
- Line 197: “meanwhileT2”
R: done
- Line 250: “Verticillium sp.”, i tis missing an end point
R: done
- Line 62: How AWC affects avocado plants? Which are the symptoms?? How the plant development and productivity are affected? In sum, how and why it is a problem for the avocado plants?
R: We agree with reviewer and information about AWC was added. Please see lines 75-86.
- Lines 76-81: Too long sentence, please rewrite.
R: done
- Line 278: 4.1, study location of which experiment? 4.3. is the same location of 4.1.? And the net house where it is?
R: We included this part and added more details to make it easier to understand. Please see lines 346-352-. In addition, we presented a scheme of the experimental phases developed under net house and field conditions. Please see lines 594-596.
- Line 289: 4.2. In which experiment and plants did you perform these analyses? In all of them?
R: done
- Line 301: Table 1 did not describe the main taxonomic characteristics analysed.
R: We included this part and added more details to make it easier to understand. Please see lines 365-366
- Lines 324-325: when did you analysed the dry weight? Also 120 days after?
R: done
- Line 356: Rootstock used in this point?
R: We included this part and added more details to make it easier to understand. Please see lines 431-441
- Line 415: Where was performed this study? The local described in 4.1?
R: We included this part and added more details to make it easier to understand. Please see lines 493-502
- Line 443: during this period in which conditions were kept the plants? They were irrigated? Differently during the drier period?
R: During this work, crop management practices different to treatments evaluated were performed according to specific indications of farmers without additional intervention of researchers (e. g., planting distances, fertilization, pest and disease management, pruning, drainage, among others). Please see lines 498-492
- Line 468: Conclusions are vague, based in your results which specific strategy would you select? e.g., type of rootstock genotype for each situation, which kind of planting tools…..
R: We agree with reviewer and information about suggestion was added. Please see lines 559-566.
Sincerely, on behalf of all authors,
Joaquin Guillermo Ramirez Gil
Ph.D. Assistant professor
Universidad Nacional de Colombia

Reviewer 2 Report
The manuscript evaluates different strategies for mitigate the adverse effects of avocado wilt complex and the climatic variability associated with ENSO phenomenon. It was selected different native avocado rootstocks according to their adaptability to drought and flooding, resistance to Phytophthora cinnamomi and Verticillium sp. infection and grafting compatibility with avocado cv. Hass. Also, the use different organic mulches were evaluated under field conditions. Adequate selection of rootstock genotypes and organic mulch decreased the susceptibility to adverse effects associated with the extreme climatic variability associated to the ENSO phenomenon and reduce the incidence and severity of avocado wilt complex disease.
The subject of the manuscript is a hot topic under the context of the prevailing and forecasted climatic conditions. In fact, urge the necessity to concentrate efforts in the search for strategies that mitigates the climate change and variability effects on agricultural productivity.
The introduction section explains well the motivation of the study exposing a clear big objective, still, it is missing information about the plant responses evaluated on the experiment, why are they important to be monitored in order to meet the proposed objectives. The introduction also fails by do not explain how the avocado wilt complex (AWC), the object of study in this manuscript, affects the plants. Moreover, no introduction is made to the proposed adaptation practices, specially the use of organic mulches.
Material and methods section should be improved, especially on the experimental design and conditions description, which are a little confuse. It is a big study, with different experiments and a lot of variables incorporated, which may have been performed in different chronological times and in different geographical locations, and with specific objectives. The reader ends up getting lost, being necessary a huge effort to understand the entire experiment. I suggest to better organize the experimental design description, for instance by an elucidative schematic representation, incorporating the different experiments and the respective conditions, the relationship between each other, and the methodologies used in each experiment to monitor the plant responses.
In the discussion section, the performance of the different selected genotypes described in 2.3, i. e. under the ENSO phenomenon, was less discussed. The conclusions are somewhat vague.
I therefore believe that the paper must be reconsidered after satisfactory major revisions.
Moreover, there are some specific comments that I would like to address:
- Regarding the formatting issues please verify carefully, I just point some examples:
Line 17: “have had”
Line 85: Grafting capital letter;
Line 97: TRI or RRI?
Figure 1 and respective caption: the acronyms did not fit. TPC and THA or RPC and TW? TV or RV?
(A) Is after the respective description, but (B) is before, please uniformize.
Line 149: “ofe”
Line 197: “meanwhileT2”
Line 250: “Verticillium sp.”, i tis missing an end point
- Line 62: How AWC affects avocado plants? Which are the symptoms?? How the plant development and productivity are affected? In sum, how and why it is a problem for the avocado plants?
- Lines 76-81: Too long sentence, please rewrite.
- Line 278: 4.1, study location of which experiment? 4.3. is the same location of 4.1.? And the net house where it is?
- Line 289: 4.2. In which experiment and plants did you perform these analyses? In all of them?
- Line 301: Table 1 did not describe the main taxonomic characteristics analysed.
- Lines 324-325: when did you analysed the dry weight? Also 120 days after?
- Line 356: Rootstock used in this point?
- Line 415: Where was performed this study? The local described in 4.1?
- Line 443: during this period in which conditions were kept the plants? They were irrigated? Differently during the drier period?
- Line 468: Conclusions are vague, based in your results which specific strategy would you select? e.g., type of rootstock genotype for each situation, which kind of planting tools…..
Author Response

(The authors gave the same response as above.)

Round 2
Reviewer 1
- I don't not seen authors' responses to each of my previous comments. I can not review and evaluate without reviewing authors' responses on how they are addressed. I am returning the manuscript.
R: All corrections were incorporated in the re-submitted document, and each consideration was considered in the response letter described in the previous paragraphs.
Reviewer 2
General comments
- The subject of the manuscript is a hot topic under the context of the prevailing and forecasted climatic conditions. In fact, urge the necessity to concentrate efforts in the search for strategies that mitigates the climate change and variability effects on agricultural productivity. The introduction section explains well the motivation of the study exposing a clear big objective, still, it is missing information about the plant responses evaluated on the experiment, why are they important to be monitored in order to meet the proposed objectives. The introduction also fails by do not explain how the avocado wilt complex (AWC), the object of study in this manuscript, affects the plants (a). Moreover, no introduction is made to the proposed adaptation practices, specially the use of organic mulches (b) .
- We agree with reviewer and information about AWC was added. Please see lines 75-86.
- We agree with reviewer and information about proposed adaptation practices were added. Please see lines 87-105.
- Material and methods section should be improved, especially on the experimental design and conditions description, which are a little confuse. It is a big study, with different experiments and a lot of variables incorporated, which may have been performed in different chronological times and in different geographical locations, and with specific objectives. The reader ends up getting lost, being necessary a huge effort to understand the entire experiment. I suggest to better organize the experimental design description, for instance by an elucidative schematic representation, incorporating the different experiments and the respective conditions, the relationship between each other, and the methodologies used in each experiment to monitor the plant responses.
R: We included this part and added more details to make it easier to understand. Please see lines 346-352-. In addition, we presented a scheme of the experimental phases developed under net house and field conditions. See lines 594-596.
- In the discussion section, the performance of the different selected genotypes described in 2.3, i. e. under the ENSO phenomenon, was less discussed. The conclusions are somewhat vague.
R: We included this part and added more details to make it easier to understand. Please see lines 281-287 and 547-554.
Specific comments
Regarding the formatting issues please verify carefully, I just point some examples:
- Line 17: “have had”
R: done
- Line 85: Grafting capital letter
R: done
- Line 97: TRI or RRI?
R: done
- Figure 1 and respective caption: the acronyms did not fit. TPC and THA or RPC and TW? TV or RV?
- Is after the respective description, but (B) is before, please uniformize.
R: done
- Line 149: “ofe”
R: done
- Line 197: “meanwhileT2”
R: done
- Line 250: “Verticillium sp.”, i tis missing an end point
R: done
- Line 62: How AWC affects avocado plants? Which are the symptoms?? How the plant development and productivity are affected? In sum, how and why it is a problem for the avocado plants?
R: We agree with reviewer and information about AWC was added. Please see lines 75-86.
- Lines 76-81: Too long sentence, please rewrite.
R: done
- Line 278: 4.1, study location of which experiment? 4.3. is the same location of 4.1.? And the net house where it is?
R: We included this part and added more details to make it easier to understand. Please see lines 346-352-. In addition, we presented a scheme of the experimental phases developed under net house and field conditions. Please see lines 594-596.
- Line 289: 4.2. In which experiment and plants did you perform these analyses? In all of them?
R: done
- Line 301: Table 1 did not describe the main taxonomic characteristics analysed.
R: We included this part and added more details to make it easier to understand. Please see lines 365-366
- Lines 324-325: when did you analysed the dry weight? Also 120 days after?
R: done
- Line 356: Rootstock used in this point?
R: We included this part and added more details to make it easier to understand. Please see lines 431-441
- Line 415: Where was performed this study? The local described in 4.1?
R: We included this part and added more details to make it easier to understand. Please see lines 493-502
- Line 443: during this period in which conditions were kept the plants? They were irrigated? Differently during the drier period?
R: During this work, crop management practices different to treatments evaluated were performed according to specific indications of farmers without additional intervention of researchers (e. g., planting distances, fertilization, pest and disease management, pruning, drainage, among others). Please see lines 498-492
- Line 468: Conclusions are vague, based in your results which specific strategy would you select? e.g., type of rootstock genotype for each situation, which kind of planting tools…..
R: We agree with reviewer and information about suggestion was added. Please see lines 559-566.
Sincerely, on behalf of all authors,
Joaquin Guillermo Ramirez Gil
Ph.D. Assistant professor
Universidad Nacional de Colombia

Round 2
Reviewer 2 Report
In general, the authors answered to the questions posed and made the suggested changes.
However, I would like to make a comment about the answers to the specific comments. The authors did not answer to most of the questions posed on the specific comments, just replying as “done” without even refer the lines in the text were the changes were made. I do not consider the correct way to reply to the queries posed, as the initial lines changed after the corrections made in the document.
Author Response
On behalf of all the authors we want to thank the reviewer for their excellent work. Based on the reviewer’s comments, we have made modifications to the original manuscript and proofread it carefully and we believe that the manuscript has been highly improved. On the other hand, We agree with reviewer abouth suggestion of added more details abouth answers to make it easier to understand.
This manuscript is a resubmission of an earlier submission. The following is a list of the peer review reports and author responses from that submission.
Round 1
Reviewer 1 Report
Ramírez-Gil et al. Plants. This study was very difficult to comprehend. I think that the authors surveyed different areas and also conducted an experiment. I feel their aim is to understand if management practices may help to moderate (lessen) disease expression that tends to increase with specific weather events.
Here are some suggestions. The list is not exhaustive.
- L2-3 “ENSO” is vague since each region is affected differently. Don’t assume readers will understand your title’s context. Can you be more specific as to the conditions (e.g. cool & wet) that increase disease expression?
- L18 “Among the major limitations” What limitations? You need a better transition between sentence one and two of abstract.
- L18 “ biotic or abiotic” Disease expression is typically a combination of biotic and abiotic conditions not one or the other.
- L20 “ the adverse effects associated with the ENSO phenomenon (El Niño-La Niña) and the most important diseases” Still not really clear what you’re talking about. I think you are wanting to say something like understand how to moderate disease outbreaks in response to ENSO events.
- L22 You list (some) experimental treatments but don’t mention your response variable(s). Also, the next sentence indicates you varied soil moisture. But this is not clear from L22. Writing has a lot of information gaps.
- L25 “ with several grafts of compatibility with the cv” This does not make sense. What is “cv” (coefficient of variation)?
- L26 “coverages” Huh?
- L26-8 “ adequate production of seedlings and planting field decrease the 27 susceptibility to adverse effects associated with the ENSO phenomenon (El Niño-La Niña) and the 28 incidence of diseases under climate variability” Does not make sense.
- L27 “ planting field decrease” Huh?
- L33 “ commonly planted commercial variety [of what?] in the internal market [what?] and for export in the world.” Wordy and confusing.
- L39-40 Unclear & confusing.
- L41 “or” (?)
- L43 Again, writing is wordy and confusing.
- L45 This content is more general than the prior paragraphs. Don’t you think you should start your introduction with the most general content and move toward specifics? You need to reorganize the content to improve flow. Also, it seems you should be focusing from the start on how crop disease(s) is impacted by climate. Not sure if climate change needs to be included or not since you seem to be focusing on ENSO. Either way, pick one and keep it is as the focus throughout.
- L46 “CO2” is directly related to disease?
- L48 “These potential impacts may be more visible in tropical regions” vague
- L49 “Colombia is a country that naturally presents a high climate variability”- wordy
- L50 “territory” but previous line you call it a country.
- L51 What changes? Vague and confusing.
- L56 what conditions? Again, the writing is very vague and lacks useful descriptions to help readers understand what is going on.
I have barely made it through the manuscript (pg 2 of 20) and already offered 20 suggestions. This is a clear indication that this manuscript is not ready to be reviewed and needs to be returned to the authors to have the writing improved. Once the writing is improved, reviewers can then critique the science.
L59 “The objective of this study was to evaluate different 59 strategies that may help to mitigate the adverse effects of climate variability and its consequences on 60 the avocado wilt complex.” Vague hypothesis. Introductions generally end with a clear hypothesis statement that is linked to some general description of the study/experiment.
L231 Was this an experiment that follows a common experimental design or a survey of fields with different properties?
L234 7x7 is an area not a distance
L317 “The treatments 317 consisted on the addition of different organic mulch:” wordy and confusing.
L378 Can you give a succinct description of the treatments here?
Reviewer 2 Report
Interesting study and interesting results. This work can be publishable if the following comments are addressed.
- Introduction: This section is very weak. Needs more details on defining the problem and previous work done related to this. No explanation of ENSO related literature review on either avocado or other comparable cropping systems was given. I encourage authors to improve Introduction overall and add more relevant literature reviews.
- Materials and Methods: Not sure why results are presented before methodology in the manuscript. I have not seen T1,T2, and T3 clearly defined in methods. Also, how did you classify ENSO years. And why only 1 year of El Nino, La Nina and Neutral was considered. If historical data available, this study needs to consider ALL available El Nino, La Nina, and Neutral years to make results and conclusions more robust. Eq.2 is so basic and not necessary.
- Results: Authors mentioned that during El Niño, root density in T3 (native rootstock) significantly (P<0.05) increased 38.0 and 87.5% at a depth of 0-30 and 30.1-50 cm, respectively, compared with the control; In contrast, during La Niña, T2 showed an outstanding increment in root density of 91.6 and 50% at a soil depth of 0-30 and 30.1-50 cm, respectively, when compared with the control (P<0.05).This is interesting results however; authors MUST explain the RATIONALE for these significant differences between El Nino, La Nina, and neutral years. Same for RRI, Incidences, and Mortality. I suggest addition another subsection in discussion to provide justification and reasoning for these statistically significant differences during ENSO events. Authors should also consider adding chart or table summarizing weather variables, temperature and precipitations, during El Nino, La Nina, and neutral years considered in this research.
- I didn’t find conclusion section.
- Miscellaneous: There are many TYPOS that needs to be cleaned up.